# Movable Platform-Based Topology Detection for a Geographic Routing Wireless Sensor Network

**DOI:** 10.3390/s20133726

**Published:** 2020-07-03

**Authors:** Runzhi Li, Jian Wang, Jiongyi Chen

**Affiliations:** College of Electronic Science and Technology, National University of Defense Technology, Changsha 410073, China; youyangxingzhu@126.com (R.L.); jiongyi_chen@126.com (J.C.)

**Keywords:** topology recovery, movable platform, location-based routing protocol

## Abstract

With the increasing adoption of the Internet-of-Things (IoT), the wireless sensors network (WSN), as an underlying application of IoT, has attracted increasing attention. Topology, the working structure used to observe WSN, is the most instinctive form in troubleshooting and has great significance to WSN management and safety. To this end, it is imperative to recover WSN topology for the purpose of network management and non-cooperative network detection. Traditional network topology recovery mainly relies on the monitoring modules installed in nodes, or an extra network attached. However, these two approaches have several limitations, such as high energy consumption for monitoring nodes, time synchronization problems, reuse failure, limitation to specific targeted networks and high cost. In this paper, we present a new approach to recover the topology of WSN that adopts location-based routing protocols, based on movable platforms. Our observation is that the network topology is consistent with the node routing, as the nodes choose the next hop according to the geological position of neighbor nodes. Hence, we calculate the cost parameters of choosing routing nodes for the targeted network according to the partial connection of the nodes. Based on those cost parameters, we can determine the topology of the whole network. More specifically, by collecting the geological position and data packets of the nodes from movable platforms, we are able to infer the topology of the WSN according to the recovered partial connection of nodes. Our approach can be easily adopted to many scenarios, especially for non-cooperative large-scale networks. The evaluation of 30 simulations shows that the accuracy of recovery is above 90%.

## 1. Introduction

With the rapid development of WSN technologies and protocols, WSN has been applied to many scenarios, such as environmental monitoring, earthquake detection, indoor positioning and battlefield monitoring. Such a large-scale deployment and application of wireless sensors has inevitably attracted more attention on the monitoring, maintenance and efficiency evaluation of networks, especially for large-scale networks. Topology, as the working structure to observe WSN, is the most instinctive form in troubleshooting and is of great significance to WSN management and safety. Therefore, the recovery of WSN topology has become a hot research topic [1], in terms of both network management and non-cooperative network detection. However, the channel conflict and link instability between WSN nodes have posted a challenge to the network monitoring and diagnosis. Moreover, the topological relation among network nodes is always changing dynamically, or is established temporarily, due to limited computing capacity, memory and power of network nodes, as well as the lack of network system structure and protocol standards [2]. At present, WSN monitoring system [3] determines network topology mainly by tracking the transport path of data packets within the network. There are two types of monitoring systems: active monitoring systems and passive monitoring systems. The active monitoring system acquires network operation information from sensor node installation and monitoring modules. Although its monitoring data are accurate and it requires no extra monitoring system, monitoring data packets occupies bandwidth, power and information transport capacity of the network itself, and lengthens the transport time delay of data packets. Passive monitoring systems require deploying WSN as the monitoring network outside the network and the network in need of monitoring is named as the target network. The monitoring network captures and analyzes the data packets of the target network without consuming any resource of the target network. Besides, failure of the target network will not affect the operation of the monitoring network. Since the WSN faces the application scenarios, and the network layer and application layer do not form a unified protocol or mainstream protocols, either the active monitoring system or passive detection system can be designed and developed only according to the specific network, which sets limits on the universality of the monitoring system to third-party networks. Moreover, the active detection system requires adding monitoring modules to each node, while the passive monitoring method requires designing and setting up an extra monitoring network of the same scale for the target network. Because of their complicated process and increased cost, both of the recovery methods cannot be applied on a large scale [4]. As described above, our paper brings forth a method to recover the topology of WSN based on a movable platform, which can recover the topology of a relatively large scale WSN by using location-based routing protocol efficiently. In the WSN using a location-based routing protocol, nodes select routing nodes according to the geological position of neighbor nodes, and the network topology is consistent with the selected node-routing. Hence, the cost parameters for the targeted network to select routing nodes are calculated according to the connection of part of the nodes, and then the topology of the whole network is calculated on the basis of cost parameters. This method needs to identify MAC frame [5] information in the data packets captured alone, determines station position code according to MAC frame information, receives nodes that are sent to base data packets as the upstream nodes among node couplings (a node coupling is defined as two nodes that can communicate directly), and determines the scope of cost parameters of the target network according to the ratio between the distance from midstream and upstream nodes to downstream nodes as well as the distance from upstream nodes to station for each node coupling. This method is characterized by good universality to target networks with different cost parameters, wide working scope, reusability, simple working steps and flexible detection range. In simulation, we detected the topology of WSNs using GEM routing protocols and GEAR routing protocols for 30 times at 256 nodes, achieving over 90% accuracy rate. Diagram of the system in this paper is as shown in Figure 1.

The rest part of this paper is arranged as below: In Section 2, we introduce the current WSN monitoring methods and relevant technologies in detail. Section 3 introduces the principles and mechanisms of the monitoring methods proposed in this paper, and elaborates their realization process. Section 4 displays and discusses all the experimental results. Section 5 summarizes the advantages, defects and restrictions of the platform, and defines the further working direction. Finally, Section 6 sums up our contributions.

## 2. Background

Monitoring on WSN can be categorized into active monitoring and passive monitoring. For active monitoring, a monitoring protocol module that obtains the specific network operation data is entailed on wireless nodes, but as monitoring flow and data flow are mixed when this method is used, higher requirements are made for the network protocol design, deployment and upgrade of the target WSN. Tools used for active monitoring in WSN include Sympathy [6], Memento [7] and Emstar [8]. There are also tools designed specifically for TinyOS [9], such as SpyGlass [10], MoteView [11], NetTopo [12], WiseObserver [13] and Octopus [14]. However, these tools will bring problems like occupation of network bandwidth, powder, information transport capacity by monitoring data, and lengthening of the transport time delay of data packets. A passive monitoring method requires a monitoring network that is deployed outside the target network, and therefore less interferes with the monitored network, as network monitoring consumes no resource of the target network. Yeo et al. [15] put forward a framework of the passive monitoring system of WLAN which deploys many monitoring nodes to capture data packets in WLAN and combine all data packets into one complete network for the purpose of tracking. Jigsaw [16] and Wit [17] improved the Yeo framework and put forward a method to track and combine, track and reason, and track the failure events of large-scale enterprise wireless network. LiveNet [18] is a passive monitoring system in WSN that tracks, combines and reasons the information lost in the rebuilding of multi-hop sensor network routing path. SNIF (Sensor Network Inspection Framework) [19] system monitors sensor networkutilizing deployed support network (DSN). SNTS (Sensor Network Troubleshooting Suite) [20] brought up one kind of statistical and data mining technology to analyze and diagnose network failure events. Pimoto [21] is similar to SNIF but their difference is that Pimoto uses Bluetooth to transfer the captured data to the gateway. The gateway is a computer that marks the data packet with a time stamp and forwards it to a central server for analysis via TCP/IP. Pimoto can also work on several sensor networks simultaneously. Its author creates a plug-in for Wireshark, a traffic analysis tool to visualize and analyze the captured data. DSN(Deployment-support networks) [22] is a toolkit for testing and monitoring WSN applications and contains a set of nodes connected to sensor nodes, all of which are connected to the server via Bluetooth. The DSN node records the events transmitted by the node application program through one of its serial interfaces and sends the information to the DSN server. In addition, they can be used as infrastructure for remote programming, command transmission and dynamic monitoring configuration. SNDS (Sensor Network Distributed Sniffer) [23] works in a way similar to SNIF or Pimoto, but it connects via Ethernet instead of Bluetooth. In NSSN [24], the sniffer node can detect the working frequency of the target WSN to be observed automatically. Via a wireless link, the collected data are sent to the monitoring server which parses, preprocesses and stores the data in the database. The information stored in the server can be visited remotely by the client which can either observe or analyze it. EPMOSt (Energy-efficient Passive Monitoring SysTem) [25] uses SNMP (Simple Network Management Protocol) proxy to collect information from the sniffer and transfer it to the server. The passive monitoring method, which requires designing and setting up an extra monitoring network at the same scale for the target network, is poorly universal and therefore cannot be applied on a large scale.

### 2.1. Common MAC Protocols

Currently researchers have brought up many MAC protocols [26] for different sensor network applications, but there is no any unified method to classify sensor network MAC protocols yet [27]. In this paper MAC protocols are roughly classified into three types: First, distributed control or central control; second, single sharing channel or multiple channels; and at last, fixed distribution channel or random visit channel. Although MAC protocols of WSN are application-oriented protocols, MAC protocols for single WSNs under each application scenario are different. For node hardware restrictions on WSN, however, we often get a chance to find the common points of all MAC protocols: First, for small power of node receiving and sending signals, when two nodes communicate, the ones within the communication range will definitely maintain their communication retreat mechanism, in order to ensure communication quality; second, as node communication environment is free, obstacles, climate or electromagnetic interference may cause packet loss during inter-code communication, handshake information mutual confirmation of information intactness when sending and receiving information becomes very necessary; third, due to small node memory and packet validity assurance, data packets shall be as small as possible and the information sending and receiving interval shall be reduced when a group of data is transported among nodes, in order to improve communication efficiency. Finally, nodes in WSN are not able to sense global information of network, and data packets will mark the codes or address of information destination node and routing node in MAC frames. In this paper, we took advantage of the common characteristics of MAC protocols, so that the platform can determine station address code (in early platform detection stage, destination node address of data packets is forcefully recognized as station address. As an increasing number of data packets are captured in the later stage, the address code that appears most frequently is station address code) and make clear data packet flow and the upstream and downstream relations of part of the nodes according to the handshake information it captures, even if the target network uses an unknown MAC protocol. Consequently, the platform’s universality to non-cooperative target networks will be greatly enhanced.

### 2.2. RSSI Algorithm

In this paper, we used RSSI (Received Signal Strength Indication) [28] to measure the distance between nodes that the platform captures. As a wireless positioning algorithm based on measurement distance, RSSI calculates the distance between unknown node and beacon node according to the signal intensity decline in its spreading process, and then calculates the coordinates of unknown node with trilateration. When indicating RSSI positioning based on the intensity of the signals received, the intensity of transmitting signal of the transmitting node is known, the receiving node will calculate the transmission loss according to the intensity of the signal received, and converts it into distance using the theoretical and empirical model, and then calculates the position of node with the existing algorithm. The general computational equation is:(1)$$P\left(d\right)=P\left(d_{0}\right)-10n\log\left(\frac{d}{d_{0}}\right)$$
where, $d_{0}$ represents the reference distance, $P\left(d_{0}\right)$ is the signal intensity corresponding to $d_{0}$, and both of them are known values. The distance *d* can be roughly calculated with the signal intensity $P\left(d\right)$ detected.

### 2.3. Space Spectrum Estimation

Spatial spectrum estimation is an important part of array signal processing. The spectrum represents the energy distribution of the signal in each frequency, while the “spatial spectrum” represents the energy distribution of the signal in each direction. Therefore, if we can get the “spatial spectrum” of signal, we can get the direction of arrival (DOA). Therefore, the spatial spectrum estimation is often called “DOA estimation” [29]. The direction finding technology of spatial spectrum estimation can achieve the simultaneous direction finding of multiple targets (including coherent signal and incoherent signal), and under the condition of low signal-to-noise ratio, the direction finding accuracy is very high, which can be fully used in the direction finding of radiation sources in complex electromagnetic environment. There are two basic algorithms, one is MUSIC (multiple signal classification) [30] algorithm. The principle of the algorithm is to decompose the autocorrelation matrix into subspaces, obtain the noise subspace and signal subspace of the matrix, and then use the orthogonal relationship between the noise subspace and the direction vector to estimate the incident direction of the signal. Another method is ESPRIT [31] (estimating signal parameter via rotational invariance techniques), which is based on the rotation invariant subspace of signal subspace class. It estimates the direction of incoming wave through the acquired signal subspace. At present, the WJ-9010 detection system (Leonardo DRS: Arlington, VA, USA) and RAF-5100 detection system (Rafael Advanced Defense Systems: Haifa, Israel) are both receivers using spatial spectrum direction finding technology. The measurement accuracy of the direction finding system can reach within 1° in the military field, and it is also used in the civil field such as auto driving. For the consideration of cost and other factors, the equipment model used in this paper is a small civil space spectrum direction finding radar, which has no interference and occlusion, so as to ensure that the accuracy of the target can reach within 10° in the range of angle. In this paper, the reference space spectrum DF radar model is shown in Figure 2.

### 2.4. Location-Based Routing Protocols

As network orienting to application scenarios, WSN uses routing protocols that differ greatly from applications. In particular, the same type of routing protocols has different details for different application scenarios. Therefore, there is not an universal routing protocol for WSN up to now. For different WSN applications, researchers brought up different routing protocols, which can be divided into four types by sensitivity of their characteristics [32]: energy aware routing, query-based routing protocol, location-based routing protocol and reliable routing protocol. We list two location-based routing protocols that are representative in terms of data receiving and sending: GEAR (Geographical and Energy Aware Routing) [33] routing protocol and GEM (Graph EMbedding for sensor networks) [34] routing protocol (illustrated in Figure 3 and Figure 4).

When GEAR [33] routing protocol is employed, it is assumed that all nodes know the position of sink, the position of neighbour node and the residual energy. Figure 3 describes detail. Propagation is divided into two stages: Firstly, sink node sends query command out to the node closest to sink within the event area according to geographical position of the area. Then the node will send the command to all nodes within the area. Monitoring data will be sent to sink along the reverse path of query information. GEAR routing protocols judge per-hop path according to actual cost and estimated cost. If the path is not built, nodes will estimate the routing cost from neighbour node to sink based on the known information and select the next node to hop to. After the path is built, the actual cost of each hop returns by the way it comes along with monitoring data. Next time, when the path is built, routing cost between nodes can be the actual cost. Cost estimation equation is expressed as:(2)c(N,R)=αd(N,R)+(1−α)e(N)

$c\left(N,R\right)$ is the estimated cost from neighbour node *N* to event area *R*, d(N,R) is the distance from node *N* to the geometric center of event area *R*, e(N) is the residual energy of node *N* and α is proportional parameter.

The basic idea of GEM [34] routing is to establish one VPCS (virtual polar coordinate system), and form a ringed tree rooted by sink to represent the actual network topology. Each node has a coordinate consisting of the hops and angle range to sink. Data transmission between nodes is completed by the ringed tree. The specific steps: sink sets its own number of hops as 0 and makes broadcast. After all nodes within one range receive it, they will set their number of hops as 1 and then make broadcast. Once receiving sub-node broadcast, sink will assign angle range for them randomly. Within 1 hop, sub-nodes seek for their sub-nodes with the same mode and assign the angle ranges they obtain. If one node receives several pieces of broadcast information at the same time, it will select the node with the strongest signal intensity as its father node, and then will form a ringed tree.

## 3. Workflow

As the technology faces specific application scenarios, WSN employs short-distance wireless communication between nodes. Due to limited node memory, power, computing capacity, single channel and weak anti-interference capacity, nodes within network do not connect each other all the time, but connect where necessary. Nodes that are not working will stay asleep. Within a short time, the platform is not able to go through all nodes to find out their connecting status. Besides, nodes should reduce the complexity of all protocols to cope with the hardware and cost. In the WSN using location-based routing protocol, nodes select routing nodes according to the geological position of neighbor nodes, and the network topology is consistent with the selected node-routing. Hence, the cost parameters for the targeted network to select routing nodes are calculated according to the connection of part of the nodes, and then the topology of the whole network is calculated on the basis of cost parameters. The method is roughly divided into the following steps:Capture node couples. The platform captures air data packets, determines the position of two nodes according to the direction and intensity of the electromagnetic wave that they send out and platform position in the process of data packet sending and receiving, and determines upstream nodes.Base address seeking. For the node couples detected, the downstream node and upstream node extension lines (defined as the direction line of node couple) should point at the station in at least one reference direction. The platform moves along the direction line of node couple and should move along the direction of new node couple after it captures node again. It will find station through changing direction line for many times.Global sampling. Upon arriving at station, the platform detects node couples and nodes of the whole network centering on the station according to the pre-determined design route.Construction of cost parameters. After the platform finishes detection, it will compare the neighbour nodes of midstream, upstream and downstream nodes according to nodes, calculate the range of cost parameter and determine the valuation of cost parameter.Reconstruction of network topology. Calculate all node couples of the whole network according to the cost parameters calculated, and re-calculate cost parameters and iterate as per the first result.Iteration and correction. Iterate the number of node hops again according to the estimated topology, and update node connection according to detection results and obtain the number of new node hops, iterate cost parameters and construct a new topology.

System structure is as shown in Figure 5.

### 3.1. Node Couple Capture

The results obtained with RSSI are greatly affected by environment in realities and the practical error might be up to 50% [35]. Therefore, we measured many times and positioned nodes with error cancellation method. We defined two nodes that communicate in network as one node couple. Supposing the movable platform is U, and it enters WSN area randomly at constant speed V, when air data packets are captured, platform U keeps moving rectilinearly for distance L, which is within the communication range of the node couple. During this period, the platform receives interaction data packets of node couples without stop, supposing the platform identifies signal source direction at precision 10–20${}^{{}^{\circ}}$ (device costs restrict signal direction recognition precision of the platform). Each time when data are captured, conversion is made with Equation (Equation 1) according to signal intensity and the intensity and distance standards that are videotaped in advance for this area, and a 10–20${}^{{}^{\circ}}$ fan area, with a radius of d, will be formed according to the position that platform U is at when it receives data packets. The value of distance d calculated will only be larger than d’, the actual distance from platform to node (during actual work, signal intensity that the platform receives is smaller than the signal intensity actually received in theory, due to such factors as obstacle, air and electromagnetic interference). Therefore, all nodes detected are within the fan area marked by the platform. When the platform is beyond the communication area, the intersection of all fan areas is roughly the position of nodes. When two positions of the node couple are acquired, it is determined that data packet type is the command sent by station or source nodes according to the MAC frame information of data packets, and the node closer to station is determined according to node receiving and sending behaviors in the node couple (shown in Figure 6).

### 3.2. Station Address Seeking

With WSN serving as the network centering on data, all nodes transfer the information they collect to station for summary. Based on this working mode, the closer a node is to station, the higher the data forwarding frequency and the more frequent node couple appears will be, which makes it possible for the platform to seek for base efficiently. On WSNs using location-based routing protocol, the direction line of node couple is always pointing at the general base direction. Moreover, the closer a node couple is to station, the stronger its directional directivity will be. Our method is to seek for station address based on this characteristic. The specific method: Before platform U captures node couple after entering network area, it moves at a constant speed according to the pre-determined trace within the network, and detects surrounding nodes. When the platform captures node couple, upstream nodes are determined according to the address code of the destination node in the MAC frame of data packet (nodes of the data packet whose receiving destination is station or nodes whose sending destination is data packets of other nodes are upstream nodes). In early platform detection stage, destination node address of data packets is forcefully recognized as station address. As an increasing number of data packets are captured in the later stage, the address code that appears most frequently is station address code. After upstream node is determined, the platform moves to upstream node quickly after it is determined and moves forward along the direction line. If no new node couple is captured, the platform will keep moving along this direction, and detect surrounding nodes. If meeting network boundary (if no data packet is captured, and no node responds to the hello information or handshake information sent out within a period of time, it is thought boundaries are met), the platform changes direction after boundaries are reflected, and then keeps moving forward along the new direction line. If data packets are captured again, the platform will change position and direction line with the above-mentioned method until it finds station. In actual work with network, routing node address in the MAC frame of data packets that the station receives is the same as destination address, and data packets that the station sends to terminals have different frequencies and protocols with the data packets inside network, helping recognize the station of non-cooperative network (shown in Figure 7).

### 3.3. Node Acquisition

If failing to capture data packets in station seeking process, the platform u will broadcast inducing signals like hello information or handshake information actively to urge nodes surrounding it to establish relationship with it. Then the platform will position nodes according to the data they send with RSSI technology. After the station is found, the platform will acquire all network nodes and node couples in rectangular spiral trace.

### 3.4. Topology Modeling

When studying two location-based routing protocols, we found that although each node has a different path to the station in the two protocols under the same scenario, the number of hops from each node to station is basically the same once energy consumption is not considered. From another perspective this phenomenon can be interpreted like this: when the number of hops from node to station is certain, routing node of GEM routing protocol should be the node that is closest to the transmitting node among all nodes of last hop, within the communication range of transmitting node; while the routing node of GEAR routing protocol should be the node that is the closest to the station among all nodes of last hop, within the communication range of transmitting node. It provides ideas for topology modeling. We constructed an universal cost equation to explain location-based routing protocol.

Supposing the transmitting node has N neighbour node (s), and the node with the minimum cost should be selected from all of them as the middle node. The cost of the *i*th neighbour code is expressed as
(3)$$D_{i}=H_{i}\times \alpha +L_{i}\times \beta +R_{i}\times \gamma$$
$D_{i}$ is the total cost of node *i*, $H_{i}$ is the number of hops of node *i* from the station, $L_{i}$ is the distance from node *i* to the station, $R_{i}$ is the distance from node *i* to source node, α,β and γ are coefficients of $H_{i},L_{i}$, and $R_{i}$. Different location-based routing protocols α,β, and γ have different proportions. To sum up, we think the topology structure of the target network can be rebuilt, as long as the proportional relationship among α,β, and γ in the cost Equation (Equation 3) of the target network is re-constructed according to the acquired information.

Supposing the platform detects M node couple (s), all of which are under the connection with minimum costs determined by target network according to Equation (Equation 3), and that the node of centering routing node in the *i*th is $g_{i}$, and minimum number of hops of each neighbour node of the transmitting node from station is λ, and there are S node(s) that are λ hop (s) from the station, then the cost of node $g_{i}$ is express as
(4)$$D_{g_{i}}=H_{g_{i}}\times \alpha +L_{g_{i}}\times \beta +R_{g_{i}}\times \gamma$$

The costs of the rest S-1 neighbour node(s) are respectively expressed as
(5)$$\begin{array}{c} D_{1}=H_{1}\times \alpha +L_{1}\times \beta +R_{1}\times \gamma \\ D_{2}=H_{2}\times \alpha +L_{2}\times \beta +R_{2}\times \gamma \\ \ldots \ldots \ldots \ldots \\ D_{s-1}=H_{s-1}\times \alpha +L_{s-1}\times \beta +R_{s-1}\times \gamma \end{array}$$
where, $H_{1}=H_{2}=\cdots =H_{S-1}=\lambda$. According to the setting that $D_{g_{i}}<D_{1},D_{g_{i}}<D_{2}\ldots \ldots D_{g_{i}}<D_{s-1}$, it can be calculated that
(6)$$\begin{array}{c} L_{g_{i}}\times \beta +R_{g_{i}}\times \gamma <L_{1}\times \beta +R_{1}\times \gamma \\ L_{g_{i}}\times \beta +R_{g_{i}}\times \gamma <L_{2}\times \beta +R_{2}\times \gamma \\ L_{g_{i}}\times \beta +R_{g_{i}}\times \gamma <L_{s-1}\times \beta +R_{s-1}\times \gamma \end{array}$$

We found from Equation (Equation 6) that the valuation range of the two coefficients β and γ cannot be calculated with the inequation, but the valuation range of γ/β can be calculated. Therefore, by taking β=1, we can obtain
(7)$$\begin{array}{c} \left(R_{gi}-R_{1}\right)\times \gamma <\left(L_{1}-L_{g_{i}}\right)\\ \left(R_{g_{i}}-R_{2}\right)\times \gamma <\left(L_{2}-L_{g_{i}}\right)\\ \ldots \ldots \ldots \\ \left(R_{g_{i}}-R_{s-1}\right)\times \gamma <\left(L_{s-1}-L_{g_{i}}\right) \end{array}$$

It can be obtained the valuation range of γ at the *i*th node couple is $\left(\gamma_{i\max},\gamma_{i\min}\right)$. Together with *M* node couples, the final valuation range γ of is $\left(\gamma_{\max},\gamma_{\min}\right)$.The smaller the valuation range is, the more node couples will be. In this paper, the midpoint $\gamma_{\mathrm{mid}\;}$ of the maximum value and the minimum value is taken as the value replacing γ, then topology rebuilding cost equation is expressed as:(8)$$D_{i}=H_{i}\times \alpha +L_{i}+R_{i}\times \gamma_{\mathrm{mid}}$$

### 3.5. Topology Rebuilding

As long as we know the number of hops from each node to the station, we can calculate all node couples again and rebuild topology with Equation (Equation 8). But the platform cannot obtain the number of hops from node to station directly. Through simulation comparison, we found different location-based routing protocols are basically in the same hops from the station at the same node and the number of hops is related to the distance from node to the station (simulation results will be introduced in detail in Part 4). According to the results, we can estimate the hops from node to the station using the distance between them after finding the station out. This method is applicable to all WSNs using location-based routing protocol.

## 4. Evaluation

### 4.1. Experiment Setup

In this paper, simulations are made with matlab software. In simulation, we did not set a specific unit of length, just 1 as the unit length. In every 1×1 grid, a node is randomly distributed. The maximum communication distance of the node is 2 (there is no super node in the network, and the communication range of the base station is 2). In practical application, the numerical conversion can be carried out in equal proportion. There are 256 nodes in the target network including the base station. Each node knows the distance between the neighbor node and itself, and the distance between the neighbor node and the base station (according to the referenced routing protocol rules). In the research, when the target network is set as the initial stage of network deployment and the network is completed, energy consumption is not the main basis for nodes to select routing nodes. In addition, in order to better simulate the actual working state of wireless sensor networks, we set up a time counter, whose range is from 1 to 19,200, and divide 19,200 times into 32 time periods on average. In each time period, 50% (adjustable) nodes are randomly selected as the source nodes to send information to the base station, and the communication process from all the source nodes randomly generated to the base station is completely simulated. In the process of communication, the sub node adopts the MAC protocol of TDMA (other MAC protocols can be simulated by adjusting the proportion of the source node and the clock of the sub node). Within the specified time range, the platform will dynamically detect the topology of the target network according to the predetermined method. In the research, the sampling speed of the platform is set to 0.5 unit length/unit time. If a pair of nodes communicate at I time, and the platform just enters the overlapping area of the communication range of the two nodes at I time, it will automatically determine that the platform captures the position of the two nodes and identifies the communication direction of the node pair. When the platform enters the base station communication area and the base station has communication, it is automatically determined as the location of the capture base station. In the whole network sampling process, one node is randomly selected as the node detected by the platform in a circle whose radius is 0.5 unit length with the platform as the center. The time counter stops after platform detection. And the target network mainly sets three routing protocols, GEAR routing protocol, GEM routing protocol and the universal routing protocol where Equation (Equation 9) serves as cost equation (according to the cost Equation (Equation 3) of GEAR routing protocol, we constructed Equation (Equation 9) as the general cost equation that considers no node energy consumption. In early network arrangement stage, most location-based routing protocols can be simulated by adjusting coefficient α and β. GEAR and GEM routing protocols are deemed as two special universal routing protocols.)
(9)$$D\left(i\right)=L\left(i\right)*\alpha +R\left(i\right)*\beta$$
D(i) is the cost of the *i*th neighbour node of source node, L(i) is the distance from node *i* to the station, R(i) is the distance from node *i* to source node, and α and β are adjusting coefficients. Energy factors are not taken into consideration by these three routing protocols. We supposed the network that the platform monitors is in its initial state and all nodes are full of energy, and there is no need to consider energy factors. The target network contains 256 nodes, each of which is randomly distributed inside each 1×1 square box, and the maximum communication distance between nodes is 2. With 50 m as the actual maximum communication distance of nodes, the target network can simulate 1600 m${}^{2}$ WSN in a real scene during the simulation. The station is located in the center of network; upon entering node couple communication range, the platform can find node couples, and can judge the communication node that is close to the station according to the MAC frame information of the data packets captured. Besides, we assumed the busyness of WSN as the ratio K between the number of source nodes that send messages to the station within a period of unit time and the total number of nodes, and network business is K = 0.5 in simulation. Supposing the platform uses unit time 1 to capture one data packet or one node, then platform speed is 0.5, corresponding to actual scene speed 12.5 m/s.

### 4.2. Setting of Prior Conditions

By comparing hops of all nodes of these three routing protocols from the station, we found 90% nodes of GEM and GEAR routing protocols are in similar hops from the station, although their nodes are selected up to different standards. We guessed for WSN using location-based routing protocol, the hops between their nodes and the station should be consistent. Through simulation, we found as long as α/β>1.5 in Equation (Equation 9), hops from the nodes of the simulated routing protocol to the station are over 90% similar with the hops from the nodes in GEM and GEAR routing protocols to the station.

Through analysis we thought in Equation (Equation 9), α/β>2 (shown in Figure 8) is the precondition that the target network selects routing nodes in minimum hops, laying a foundation for predicting how many hops the nodes of location-based routing protocol are from the station. According to the simulation, we can obtain the target network that uses location-based routing protocols and same nodes are in similar hops from the station. Although the hops from a node to the station change due to lots of influencing factors, the main factor is the distance from node to station. In view of the above, we can know it applies to similar target networks to estimate node hops by taking advantage of the distance from node to station. In terms of fitting node hops, we found the distance from node to station is larger than node hops, but the distance, after quantified by the maximum communication distance of node (referring to as quantified distance from node to station) is always smaller than node hops (shown in Figure 9). Based on these two estimation values, we rounded off and averaged numbers again and again with the method of bisection, and finally fitted one estimation curve of node hops, which is roughly consistent with the actual curve of node hops (shown in Figure 10).

### 4.3. Accuracy Measurement

#### 4.3.1. Accuracy of Topology

After obtaining the estimated hops from each node to the station, cost parameters are calculated with cost Equation (Equation 3) and node couples are established according to them within the whole network. While calculating parameters, we found that the valuation range of cost parameters that are obtained using the estimated node hops is always self-contradictory, (for example, the valuation range of γ at the *i*th node couple captured is $\left(\gamma_{i\max},\gamma_{i\min}\right)$, the valuation range obtained at the *j*th node captured is $\left(\gamma_{i\max},\gamma_{i\min}\right)$), but $\left(\gamma_{i\max},\gamma_{i\min}\right)$. For this reason, we introduced one valuation compensation method. We thought, although the valuation range of cost parameters cannot be determined due to the error in estimating node hops, the general proportion of the upper and lower limits will not be changed. Therefore, we no longer used the method of calculating the valuation range of γ using each node couple before calculating the minimum range while calculating cost parameters, but did calculate the valuation range of γ for each code. As only the unilateral range of γ can be calculated with single nodes, the other side is made up with the maximum and minimum thresholds, the absolute values of which are the same. At last, all valuation ranges are averaged to obtain the valuation range $\left(\gamma_{i\max},\gamma_{i\min}\right)$ of γ, and then the median $\gamma_{\mathrm{mid}}$ is calculated and $e^{\gamma_{\mathrm{mid}}}$ is taken as the cost parameter. According to the simulation results, over 50% correction rate can be achieved by rebuilding topology structure for the first time. After rebuilding the topology for the first time, the results of the first formation is compared with the node couples acquired, node couples that are different from correction and acquisition results are corrected, and the minimum hops that each node can obtain are adjusted again for the corrected results. Besides, costs are calculated and node couples are selected again according to new hops. 255 node pairs detected by the platform are compared with the actual node pairs, and the correct ratio between the total number of node pairs and the total number of node pairs is used as the reference of accuracy. After three times of iteration, the accuracy rate of the rebuilt topology is more than 95% (the comparison of rebuilt topology and original topology is shown in Figure 11 and Figure 12; the accuracy of topology reconstruction is shown in Figure 13).

#### 4.3.2. Accuracy of Node Positioning

In the platform, RSSI is used to measure the distance from the node, and spatial spectrum direction finding technology is used to get the node direction. In order to reduce the error, we adopt the idea of multiple measurements to get the intersection. In the simulation experiment, the platform uses irregular curve track motion to locate the nodes from the near end area, the middle area and the far end area of the node communication range. For each region, the platform has carried out 30 localization experiments on the target node independently. In each experiment, the initial position of the platform, the motion route of the platform, the sampling times, the position of the target node, the RSSI measurement error of the platform and the direction finding accuracy of the spatial spectrum are all random changes. In the motion of the platform, the electromagnetic wave of the signal source is received in equal time interval. When the platform is closer to the node, the more accurate the RSSI ranging is, the more accurate the spatial spectrum direction finding is. Otherwise, the accuracy decreases. The closer the platform is to the node, the more times it receives electromagnetic wave, and vice versa. Based on the simulation results, even if the platform is in the remote area, the ratio of the positioning error to the communication range is within 10% (the illustration of detection is shown in Figure 14 and Figure 15; the accuracy of detection is shown in Figure 16).

### 4.4. Performance Measurement

It takes about 300 unit times for the platform to seek for station, and no more than 1400 unit times to complete global detection (shown in Figure 17). The platform is able to acquire about 40 node couples (shown in Figure 18), and the time it uses increases gradually as the scale of the target network becomes larger.

Besides, another factor that affects modeling accuracy rate of the platform is the quantity of node couples captured. The more the node couples captured, the higher the modeling accuracy rate. A busier and larger target network brings higher platform accuracy. Thirdly, in consideration of the difference in node hardware and the environment and consumption of battery in target network, the topology of the target network changes at all times and irregularly. In the current stage, the platform is only applicable to target networks with a fixed network structure or at the preliminary stage of network with sufficient charge. At last, the platform is highly universal and in particular, brings good results for non-cooperative networks. The platform requires no tedious monitoring or network and is recyclable and reusable.

## 5. Discussion

For systems that monitor the whole WSNs with a single movable platform, their technical development is hindered in the several aspects: Firstly, low efficiency. Due to hardware and energy consumption, sensor nodes can only transmit signals within tens of meters to hundreds of meters at the same frequency and usually use CSMA/CA (carrier sense multiple access with collision avoidance) protocol. So when one node couple communicates, other nodes within its communication range will listen or fall asleep. Even if broadcast information is sent and communicates with surrounding nodes, only one node can be detected. But the sensing would fail if there are sleeping nodes. In theory, as long as time T that is large enough is given, the platform can go through all nodes within the network, and we can always detect all path information for the network in fixed routing path. However, as network keeps expanding, especially for network models with over 1000 orders of magnitude, it takes a long time for this method to work, and network routing path would change, resulting in invalid detection results. Secondly, incomplete detection information. Compared with traditional spot monitoring system, single platform is not able to obtain data of the same moment within the whole network or the whole-process path information of data packets but is only able to infer according to the models built. Thirdly, poor flexibility. The platform uses the method of reproducing the whole path of network using the known path information but such method is only applicable to target networks using location-based routing protocols at present. Lastly, this method cannot achieve good results without constant iteration, and too many iterations will cause overload. Next, we will study how to prevent overload, and bring up effective methods to detect the topology of WSNs in clustering topology structure.

## 6. Conclusions

This paper brings forth a method to recover the topology of WSN (wireless sensors network) based on a movable platform (only applicable to WSNs using location-based routing protocol). It is superior to the previous work thanks to its simple arrangement, reusability, higher accuracy of larger scale and applicability to non-cooperative network topology. This method captures data packets to recover the position of target network node couples, seeks for network station through data packet transmission direction and completes acquiring nodes and data packets of the whole network based on clarified station position; constructs cost equations and calculates cost parameters using the detected node couples, and infers network topology reversely. At the same time, in the process of iterating the cost parameters, the error of the platform in the sampling will also be reduced with the iteration. Among the 30 independent simulation experiments, the platform can complete rebuilding topology within a short time with accuracy rate above 90%.

## Figures and Tables

**Figure 1 sensors-20-03726-f001:**
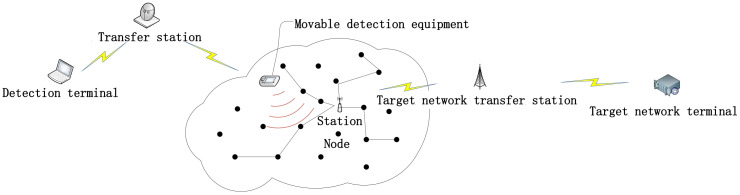
System Illustration.

**Figure 2 sensors-20-03726-f002:**
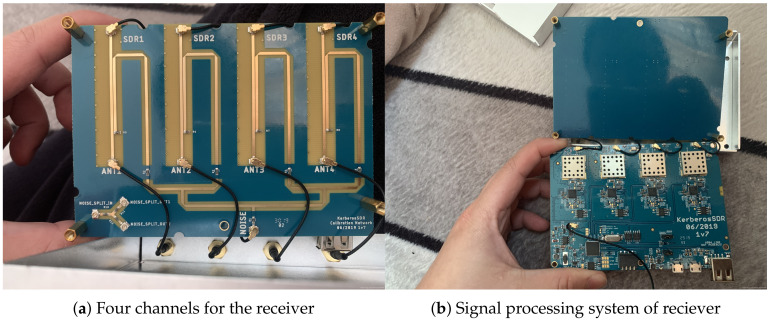
Internals of the spatial spectrum direction finding radar.

**Figure 3 sensors-20-03726-f003:**
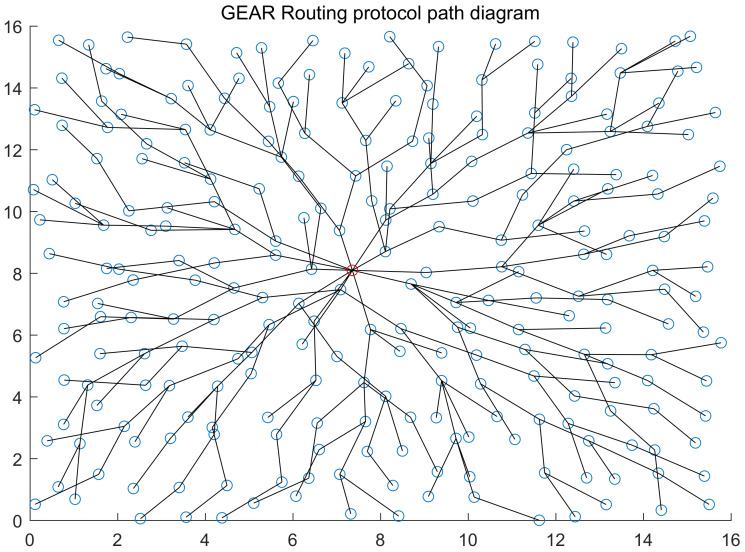
The figure shows the WSN topology corresponding to GEAR routing protocol without energy consumption factors taken into consideration. The red nod represents station. There are a total of 256 nodes within the network, the maximum communication distance of nodes is 2, and 1 node is randomly distributed within each 1×1 square box.

**Figure 4 sensors-20-03726-f004:**
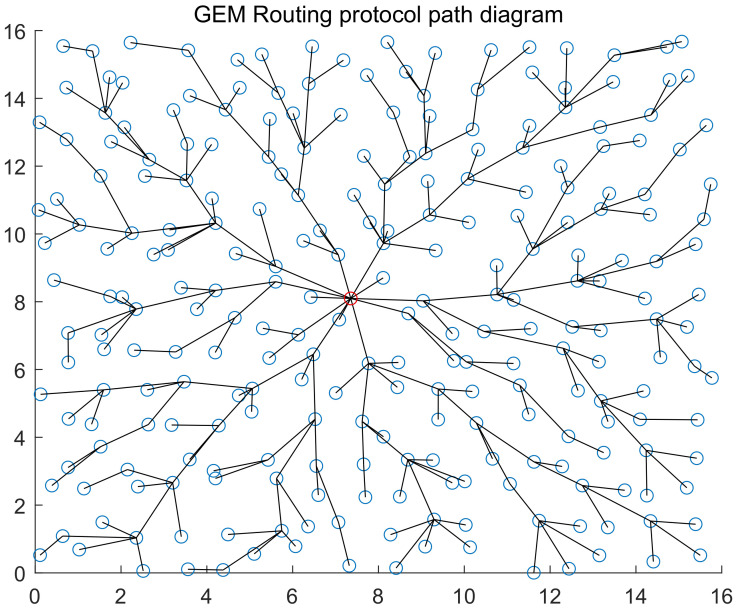
The figure shows the topology diagram of the WSN corresponding to GEM routing protocol. The red nod represents station. There are totally 256 nodes within the network, the maximum communication distance of nodes is 2, and 1 node is randomly distributed within each 1×1 square box.

**Figure 5 sensors-20-03726-f005:**
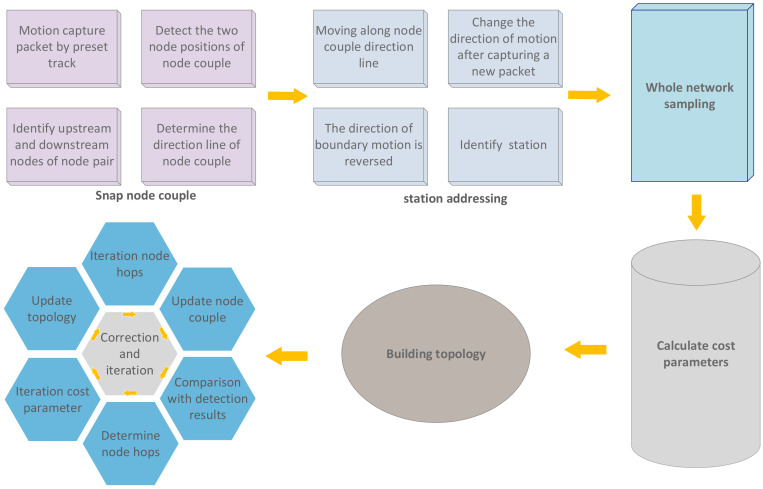
System Structure.

**Figure 6 sensors-20-03726-f006:**
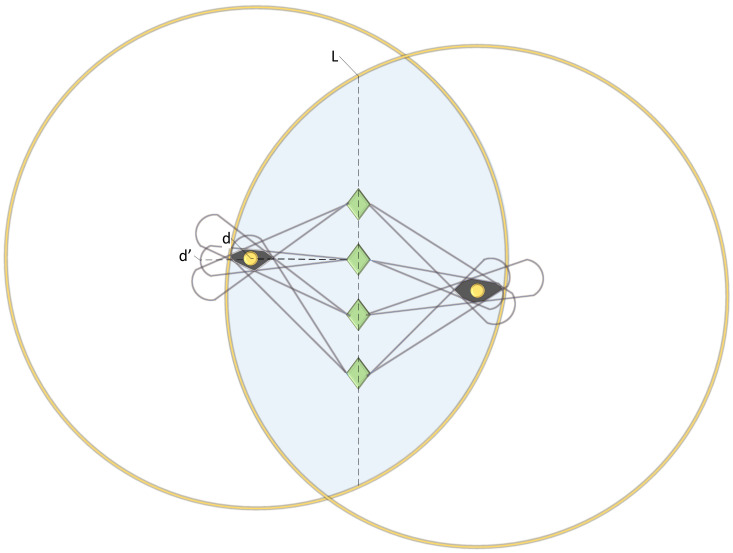
How the platform captures data packets and determines node couple.

**Figure 7 sensors-20-03726-f007:**
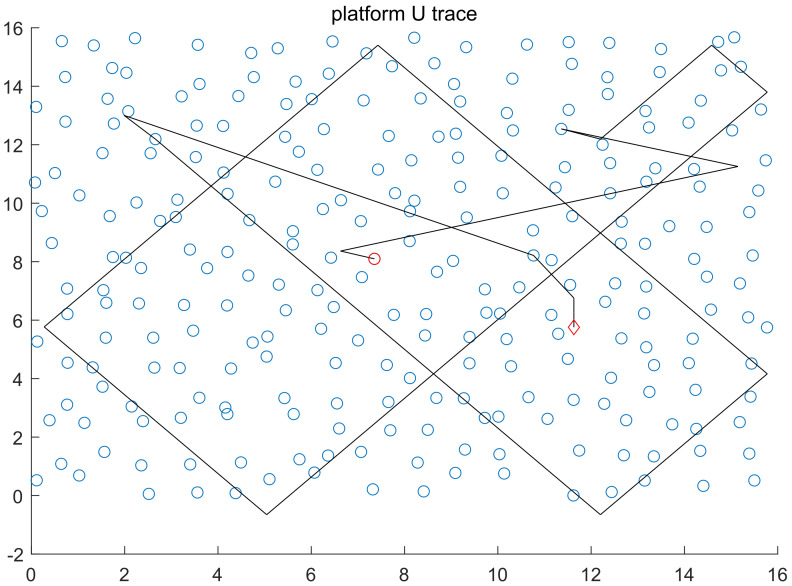
The figure is the trace diagram about how the platform seeks for station based on above rules in 1 time of simulation, where the red circle represents station position and the red rhombus is the starting position of platform (randomly), and the black lines represent the motion trace of the platform when it seeks for address. After entering the target network randomly, the platform in the diagram will detect according to the pre-determined trace (the pre-determined trace is not unique). Even if the upstream node is judged wrong in the early stage, there will be one dimension facing the station after boundary reflection, and reaching station node after many times of boundary reflection. (Platform motion trace is marked out for the purpose of better explaining how the platform seeks for station).

**Figure 8 sensors-20-03726-f008:**
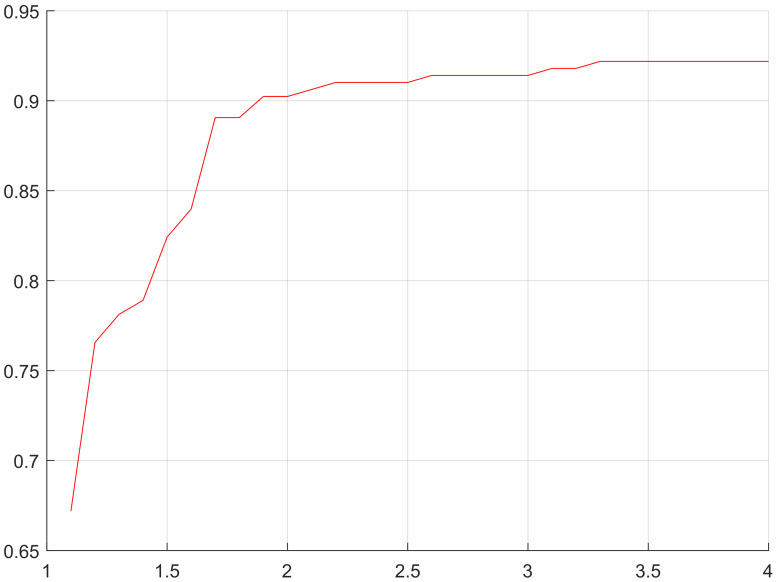
*x*-coordinate is the value of α/β and the *y*-coordinate is the proportion of all nodes of the simulated routing protocol and GEM routing protocol that are in same hops from the station. When α/β = 2, the proportion is larger than 90%.

**Figure 9 sensors-20-03726-f009:**
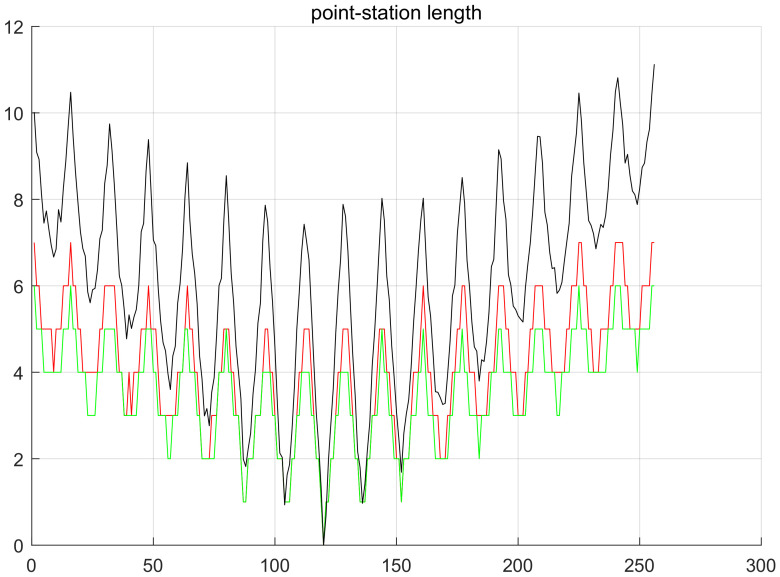
*x*-coordinate is the codes of all nodes, and the 120th node is the station. *y*-coordinate is the hops or distance of nodes from the station. The black curve is the distance curve from nodes to the station, the red curve is the universal node hop curve, and the green curve is the quantified distance curve from nodes to the station.

**Figure 10 sensors-20-03726-f010:**
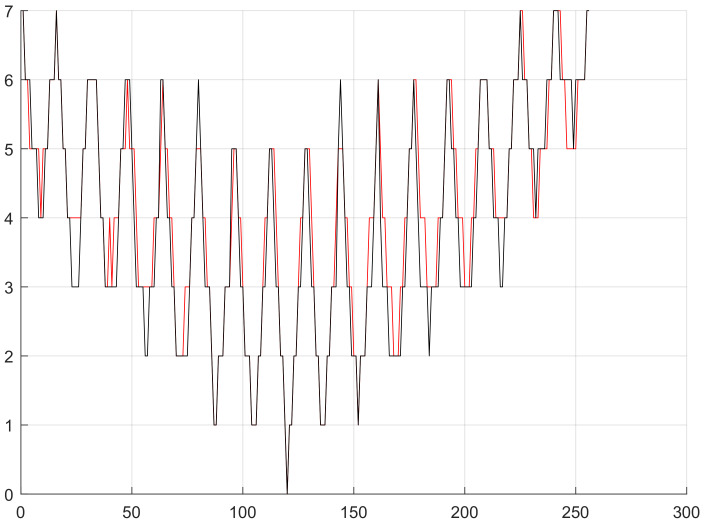
*x*-coordinate is the codes of all nodes, and the 120th node is the station. *y*-coordinate is the hops or distance from nodes to the station. The black curve is the estimated node hop curve after fitting and the red curve is the universal node hops.

**Figure 11 sensors-20-03726-f011:**
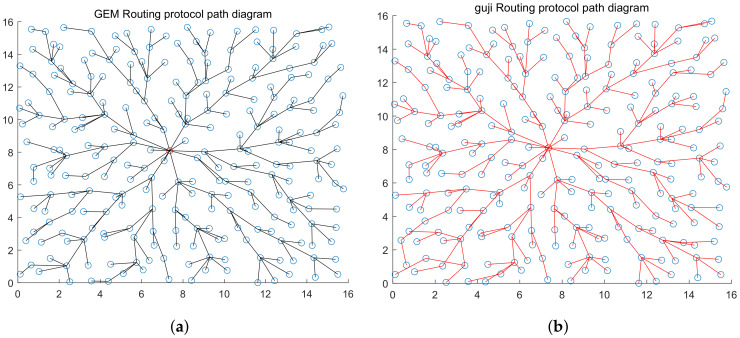
Part (**a**) is the original topology of the target network of GEM routing protocol, and Part (**b**) is the topology rebuilt by the platform.

**Figure 12 sensors-20-03726-f012:**
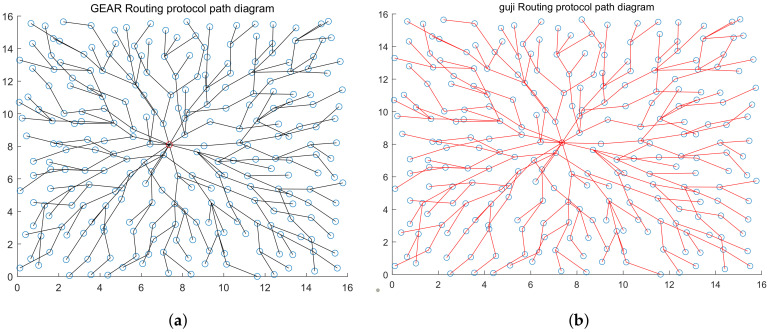
Part (**a**) is the original topology of the target network of GEAR routing protocol, and Part (**b**) is the topology rebuilt by the platform.

**Figure 13 sensors-20-03726-f013:**
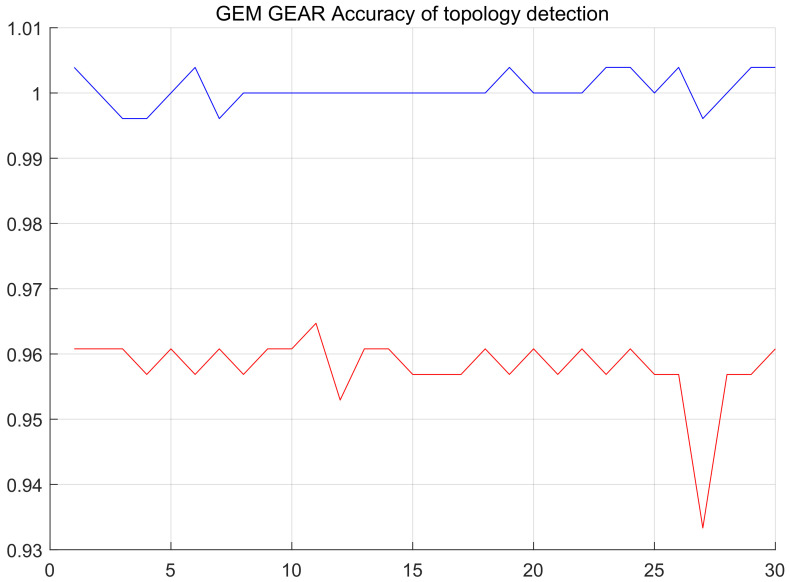
The red lines represent the topology detection accuracy rate of the target network using GEM routing protocol in 30 experiments. The blue lines represent the topology detection accuracy rate of the target network using GEAR routing protocol in 30 experiments.

**Figure 14 sensors-20-03726-f014:**
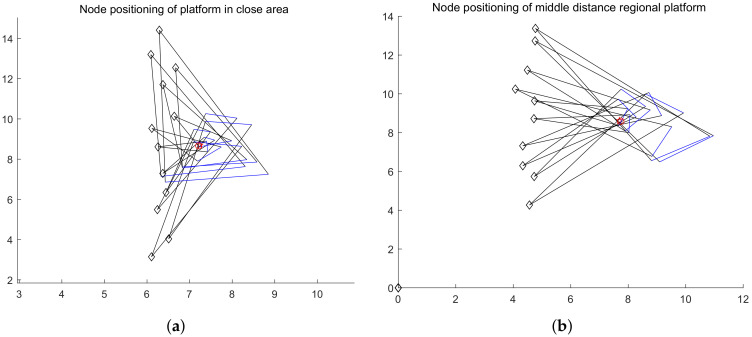
(**a**,**b**) is the positioning simulation of the platform node when the communication range is near and middle. (☆ is the positioning position of the platform to the node, and ◯ is the actual position of the node).

**Figure 15 sensors-20-03726-f015:**
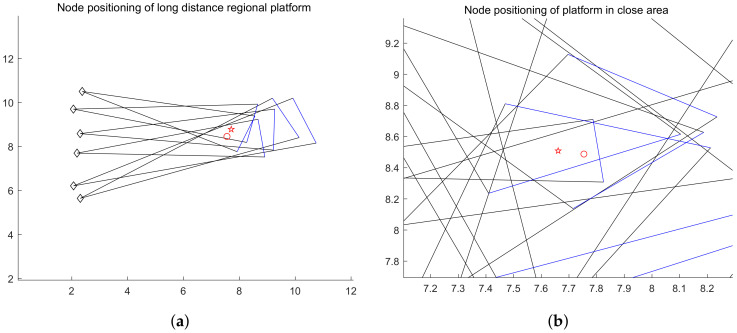
(**a**) is the positioning simulation of the platform node when the communication range is far; (**b**) is the simulation precision display. (☆ is the positioning position of the platform to the node, and ◯ is the actual position of the node).

**Figure 16 sensors-20-03726-f016:**
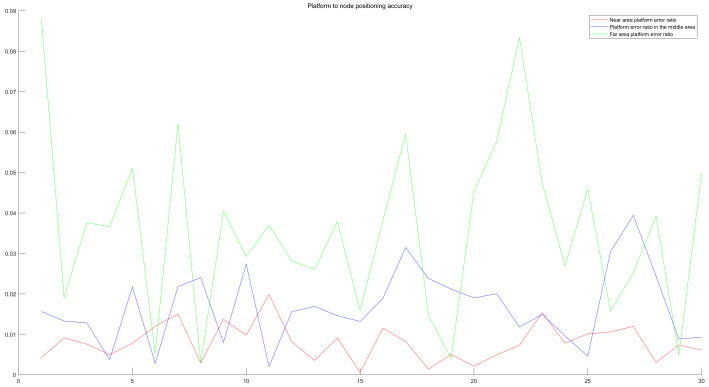
The platform’s error measurement of node positioning in the near, middle and far communication range.

**Figure 17 sensors-20-03726-f017:**
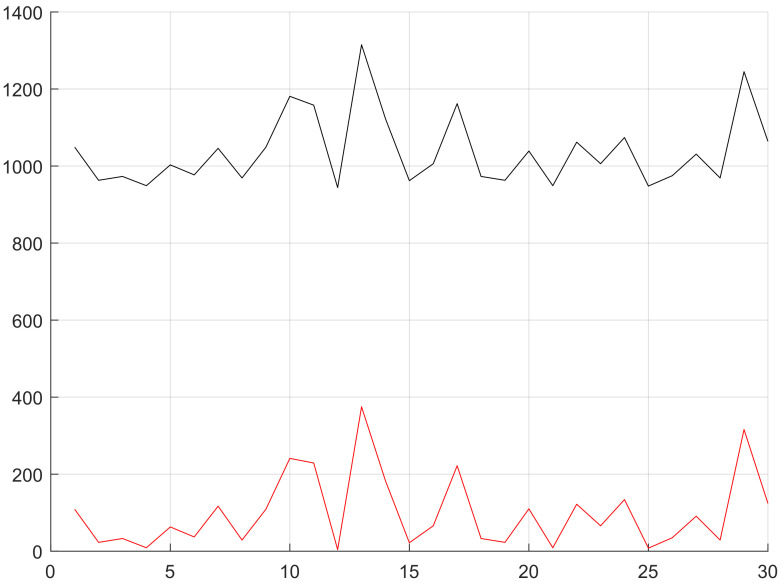
The black curve is the time that the platform takes to complete all sampling, and the red curve represents the time it takes to find the station.

**Figure 18 sensors-20-03726-f018:**
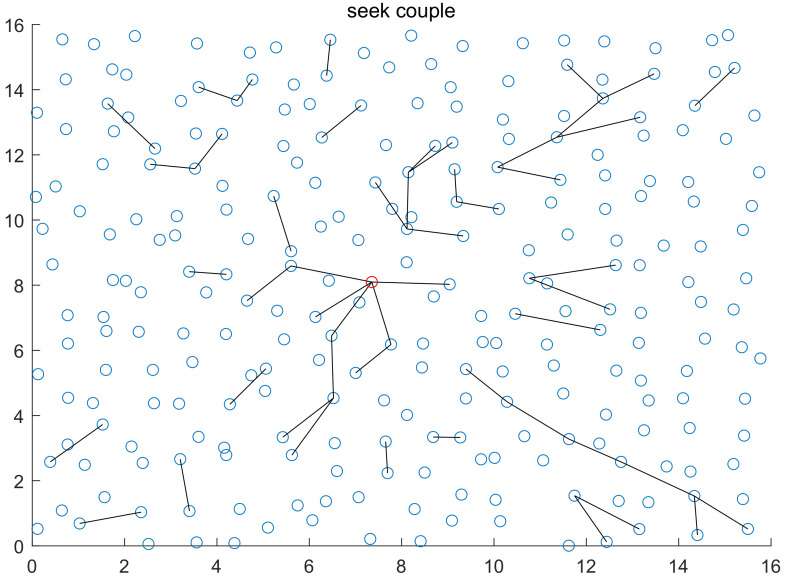
Node Couples the Platform Detects after Finishing Sampling Completely.

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
