# Peer review of "Movable Platform-Based Topology Detection for a Geographic Routing Wireless Sensor Network"

_sensors, 2020, doi:10.3390/s20133726_

Round 1

Reviewer 1 Report

The authors propose a mechanism to guess the topology of a WSN (under specific conditions and using specific routing protocols) by using a mobile platform. 

In my opinion the title of the paper should be changed to highlight the fact that the method isn't valid to detect the topology of any WSN, but only of very specific ones (a network using certain location-based routing protocols). 

Along the paper the authors have made quite bold assumptions such as that RSSI can be used to calculate distance to an unknown node (it can be used to estimate it, but with certain error and if you have characterised beforehand the scenario and the nodes), and even worse, that you can determine the "angle" from which a signal arrives (while you measure RSSI also). There are also other assumptions such as the platform travels at a constant speed and in a rectilinear way and so on. 

Since these assumptions are quite unrealistic, it is highly unfeasible that this method could be applied in a real WSN. The results and claims presented by the authors are therefore backed only by simulation tests, but could not be tested in a real scenario.

Given these facts I think the author should at least:

  • Better describe the simulation model they have used. They mention the simulation tool (MATLAB), but they do not discuss their simulation model at all. Is it a discrete-event simulator or a time-slice simulator? Which protocols layer are considered and modelled? Which are the parameters?
  • Discuss how the errors in distance estimation using RSSI and the unfeasibility of determining the direction of a signal, and so on, would affect their method performance in a more realistic simulation.

Thank you and regards

Author Response

Dear Reviewer :

    Thanks to your guidance, we have sorted out and answered all the comments and wrote them in the attachment and the updated paper. Please check the attachment for details.

Reviewer 2 Report

This paper presents a method to recover the topology of WSN based on movable platform. There are some major issues.

In the evaluation part, authors should compare the performance with related works in terms of  accuracy.

And the experiment setups are not well detailed in simulation model, methodology, and environments.

Authors concluded the accuracy rate of the rebuilt topology is more than 95%, but they did not explain how the value was calculated.

Author Response

(The authors gave the same response as above.)

Round 2

Reviewer 1 Report

The authors have correctly addressed the reviewers' comments, so I think it is suitable for publication now.

Reviewer 2 Report

I recommend this paper for publication in present form.